



# How non-equilibrium aerosol chemistry impacts particle acidity: the GMXe AERosol CHEMistry (GMXe–AERCHEM, v1.0) sub-submodel of MESSy

Simon Rosanka[1,2], Holger Tost[3], Rolf Sander[4], Patrick Jöckel[5], Astrid Kerkweg[1,6], and Domenico Taraborrelli[1,6]

[1]Institute of Energy and Climate Research: Troposphere (IEK-8), Forschungszentrum Jülich GmbH, Jülich, Germany
[2]Department of Chemistry, University of California, Irvine, California, United States
[3]Institute for Physics of the Atmosphere, Johannes Gutenberg University Mainz, Mainz, Germany
[4]Atmospheric Chemistry Department, Max Planck Institute for Chemistry, Mainz, Germany
[5]Deutsches Zentrum für Luft- und Raumfahrt (DLR), Institut für Physik der Atmosphäre, Oberpfaffenhofen, Germany
[6]Center for Advanced Simulation and Analytics (CASA), Forschungszentrum Jülich, Jülich, Germany

**Correspondence:** Simon Rosanka (s.rosanka@fz-juelich.de, srosanka@uci.edu)

**Abstract.** Aqueous-phase chemical processes in clouds, fog, and deliquescent aerosols are known to alter atmospheric composition and acidity significantly. Traditionally, global and regional models predict aerosol composition by relying on thermodynamic equilibrium models and neglect non-equilibrium processes. Here, we present the AERosol CHEMistry (GMXe–AERCHEM, v1.0) sub-submodel developed for the Modular Earth Submodel System (MESSy) as an add-on to the thermo-

dynamic equilibrium model (i.e., ISORROPIA-II) used by MESSy's Global Modal-aerosol eXtension (GMXe) submodel. AERCHEM allows the representation of non-equilibrium aqueous-phase chemistry of varying complexity in deliquescent fine aerosols. We perform a global simulation for the year 2010 by using the available detailed kinetic model for the chemistry of inorganic and small oxygenated organics. We evaluate AERCHEM's performance by comparing the simulated concentrations of sulfate, nitrate, ammonium, and chloride to in situ measurements of three monitoring networks. Overall, AERCHEM re-

produces observed concentrations reasonably well. We find that especially in the USA, the consideration of non-equilibrium chemistry in deliquescent aerosols reduces the model bias for sulfate, nitrate, and ammonium, when compared to simulated concentrations by ISORROPIA-II. Over most continental regions, fine aerosol acidity simulated by AERCHEM is similar to the predictions by ISORROPIA-II but tends to simulate slightly lower aerosol acidity in most regions. The consideration of non-equilibrium chemistry in deliquescent aerosols leads to a significant higher aerosol acidity in the marine boundary layer,

which is in line with observations and recent literature. AERCHEM allows investigating the global-scale impact of aerosol non-equilibrium chemistry on atmospheric composition. This will aid the exploration of key multiphase processes and improve the model predictions for oxidation capacity and aerosols in the troposphere.



## 1 Introduction

Aqueous-phase chemical processes in clouds, fogs, and deliquescent aerosols are known to alter atmospheric composition sig-
nificantly and produce species that can not be formed in the gas phase (Ervens, 2015). In addition, multiphase processes are
known to produce aqueous-phase secondary organic aerosols (aqSOA) from biogenic and anthropogenic volatile organic com-
pounds (VOCs) (Carlton et al., 2008). Aerosol acidity influences the lifetime of pollutants, ecosystem health and productivity,
Earth's climate, and human health. In general, the acidity of condensed phases in the atmosphere is controlled by low volatile
gases (e.g., $H_2SO_4$), semivolatile gases (e.g., HCl, $NH_3$, and $HNO_3$) as well as organic acids. Mainly driven by different
water content, the acidity (defined as pH) of condensed phases in the atmosphere typically ranges for deliquescent aerosols
from -1 to 5, for clouds and fog from 2 to 7, and from 3 to 7 for rain droplets (Pye et al., 2020). Anthropogenic emissions like
ammonia ($NH_3$) are known to reduce acidity, whereas others like nitrogen oxides ($NO_x$), sulfur dioxide ($SO_2$), and organic
acids (e.g., formic acid, HCOOH) increase acidity. Recently, atmospheric aerosols have received attention since they have
direct implications for air quality, aerosol toxicity and thus human health, cloud formation and thus climate by altering aerosol
hygroscopicity, and ecosystems via acid deposition and nutrient availability. A realistic prediction of aerosol composition and
thus aerosol acidity in atmospheric chemistry models, is thus crucial to tackle current and future challenges.

Traditionally, regional and global models calculate aerosol composition by using a thermodynamic equilibrium model. These
thermodynamic models are mainly limited to a few low and semivolatile inorganic gases and neglect organic acids. However,
some models also include the reactive uptake onto aerosols of a selection of chemical compounds. The representation of
non-equilibrium aqueous-phase chemistry is mainly limited to cloud droplets and significantly differs in the degree of com-
plexity (Ervens, 2015). Recently, Rosanka et al. (2021c) developed a very detailed aqueous-phase chemical mechanism suitable
for global model applications, finding significant implications for the abundance of oxygenated volatile organic compounds
(OVOCs) and tropospheric ozone ($O_3$) (Rosanka et al., 2021b). Further, Franco et al. (2021) demonstrated the importance
of aqueous-phase processes to properly represent the atmospheric abundance of formic acid. Past attempts to globally repre-
sent non-equilibrium chemistry in deliquescent aerosols were hindered by numerical issues and mostly limited to the marine
boundary layer (Kerkweg et al., 2007). In order to overcome this modelling limitation, we develop the AERosol CHEMistry
(GMXe–AERCHEM, v1.0) sub-submodel as an add-on to the thermodynamic equilibrium model (i.e., ISORROPIA-II) of the
Global Modal-aerosol eXtension (GMXe; Pringle et al., 2010) submodel in the Modular Earth Submodel System (MESSy
version 2.55.0; Jöckel et al., 2010). It allows representing non-equilibrium aqueous-phase chemistry of varying complexity in
the deliquescent phase of accumulation and coarse aerosols. This study presents a short overview on the current representa-
tion of aerosols in MESSy (Sect. 2), AERCHEM's technical development (Sect. 3), a first evaluation of the simulated aerosol
composition and acidity (Sect. 4), a discussion of model limitations (Sect. 5), and future application scenarios (Sect. 6).

## 2 Aerosol representation in MESSy

MESSy is a numerical chemistry and climate simulation system that includes submodels describing tropospheric and middle
atmospheric processes and their interaction with oceans, land, and human influences (Jöckel et al., 2010). MESSy contains





various representations of aerosols and aerosol related processes described by Jöckel et al. (PTRAC; 2008), Kaiser et al. (MADE3; 2019), and Pringle et al. (GMXe; 2010). However, in the following, we focus on the submodels used for this study. The following section provides a brief overview on the representation of aerosols and related processes in MESSy, with a focus on properties important to represent non-equilibrium aqueous-phase chemistry in deliquescent aerosols.

## 2.1 Chemical processes in MESSy

In most atmospheric chemistry models, multiphase chemistry is represented as a system of coupled ordinary differential equations (ODE). Ideally, gas-phase and aqueous-phase processes in clouds and aerosols would be integrated in a single ODE system. However, this will result in a very large and stiff ODE system, which is numerically hard to solve (Sandu et al., 1997). In order to improve numerical efficiency, chemical processes in MESSy are calculated separately for the cloud, aerosol, and gas-phase in sequence (operator splitting framework). Figure 1 illustrates the order in which these chemical processes are executed in MESSy. In a first step, the SCAVenging submodel (SCAV; Tost et al., 2006) is used to simulate the removal of trace gases and aerosol particles by clouds and precipitation. SCAV calculates the transfer of species into and out of rain and cloud droplets using the Henry's law equilibrium, acid dissociation equilibria, oxidation reactions, and aqueous-phase photolysis reactions. Afterward, all aerosol processes are calculated by GMXe (see Sect. 2.2). Lastly, the Module Efficiently Calculating the Chemistry of the Atmosphere (MECCA, Sander et al., 2019) is used to calculate gas phase chemistry.

## 2.2 The Global Modal-aerosol eXtension (GMXe)

GMXe is used to calculate aerosol microphysics using seven modes to describe the log-normal size distributions. Three hydrophobic modes that cover the size spectra of Aitken, accumulation, and coarse modes and four hydrophilic modes that cover the same size range and additionally the size spectrum of nucleation. Each mode is defined in terms of total number concentration, particle mean radius, and geometric standard deviation of the radius distribution. Within each size mode, the aerosol composition is internally mixed (uniform) but varies between modes (externally mixed). Table 1 provides a summary of the recommended GMXe submodel setup for each mode, when using AERCHEM.

ISORROPIA-II is used to calculate the thermodynamic equilibrium, which calculates the gas/liquid/solid equilibrium partitioning of $K^+$–$Ca^{2+}$–$Mg^{2+}$–$NH_4^+$–$Na^+$–$SO_4^{2-}$–$NO_3^-$–$Cl^-$–$H_2O$ aerosols. For this, it considers 19 salts in the solid phase and 15 aqueous-phase compounds. When using AERCHEM, it is assumed that all aerosols are in a metastable state, meaning that all aerosols have an aqueous phase which allows for supersaturation of dissolved salts. A detailed description of all processes represented in GMXe and ISORROPIA-II is provided by Pringle et al. (2010) and Fountoukis and Nenes (2007), respectively.

## 2.3 Aerosol water

The representation of non-equilibrium aerosol chemistry is inherently dependent on the aerosol liquid water content. In GMXe it is assumed that each particle mode is internally mixed but ISORROPIA-II only considers the uptake of water by inorganic





compounds ($W_{\text{inorganic}}$, $\text{g m}^{-3}$). The aerosol water due to organic compounds, which is added to the aerosol water predicted by ISORROPIA-II, is calculated based on the mass concentration ($m_{\text{s}}$, $\text{g m}^{-3}$) of all organics dissolved, the water ($\rho_{\text{w}}$, $\text{g m}^{-3}$) and organic aerosol ($\rho_{\text{s}}$, $\text{g m}^{-3}$) density, the relative humidity ($\text{RH}, 0-1$), and the hygroscopicity parameter ($\kappa_{\text{organic}}$) of the soluble organic:

$$W_{organic} = m_s \cdot \frac{\rho_w}{\rho_s} \cdot \frac{\kappa_{organic}}{\left(\frac{1}{RH} - 1\right)} \tag{1}$$

The organic aerosol (OA) composition and evolution in the atmosphere is simulated within GMXe. Primary emitted organic aerosol are mainly emitted into the hydrophobic Aitken mode, with only a small fraction being assumed to be directly soluble and emitted into the hydrophilic Aitken mode. Here, an initial hygroscopicity parameter of 0.1 is assumed, as suggested by Lambe et al. (2011). GMXe represents the formation of secondary organic aerosols (SOA) from isoprene, $\alpha$-pinene, $\beta$-pinene, toluene, and xylene. For this, an additional SOA model was implemented into GMXe based on the two-product model originally proposed by Odum et al. (1996). This model has been described in detail elsewhere (Tsigaridis and Kanakidou, 2003; Zhang et al., 2007; O'Donnell et al., 2011) and a general description is presented in the supplement of this manuscript. A summary of all hygroscopicity parameters used for each SOA species is provided in Table S2.

## 2.4 Cloud-aerosol interactions

Similarly to gas-phase species, aerosols are directly influenced by scavenging processes, which are represented by the submodel SCAV in MESSy. First, SCAV computes the fraction of nucleation scavenging for each aerosol species. The scavenged fraction of each aerosol species is assumed to be instantly activated and represents the initial concentrations in cloud droplets used to compute in-cloud chemistry. Subsequently, SCAV calculates cloud chemical processes based on an aqueous-phase chemical mechanism selected by the user. While processing chemical processes in the aerosol and gas-phase, it is assumed that the cloud composition remains constant and all cloud species reside within cloud droplets. After GMEx and MECCA have calculated all aerosol processes and gas-phase chemistry, respectively, the cloud composition is considered to reside in the coarse mode if the cloud evaporates.

## 2.5 Additional aerosol removal processes

In addition to aerosol scavenging, the removal of aerosol tracers by dry deposition and sedimentation is considered by using MESSy's Dry DEPosition (DDEP) and SEDImentation (SEDI) submodel, respectively. From a technical point of view, dry deposition is only applied in the lowest model layer, whereas sedimentation occurs in the entire vertical column. In the case of aerosol particles, sedimentation is a significant sink, but is no sink for trace gases. A detailed description of the technical representation in MESSy of both processes is presented by Kerkweg et al. (2006a).



## 3 The AERCHEM sub-submodel

### 3.1 Integration of AERCHEM in GMXe

AERCHEM is developed as an add-on to the thermodynamic equilibrium model (i.e., ISORROPIA-II) of GMXe. Similar to MESSy, the sequence of simulated aerosol processes in GMXe are ordered by their expected timescale within the atmosphere. The thermodynamic equilibrium is expected to be reached quickly, whereas the non-equilibrium aerosol chemistry is expected to act on longer time scales. Thus, AERCHEM is executed in series after the thermodynamic equilibrium calculations performed by ISORROPIA-II (see Fig. 1).

### 3.2 Representation of phase transfer

In AERCHEM, the exchange rate coefficients are calculated before the integration of the ODE system following Schwartz (1986). The forward ($k_{ex}^{f}$) exchange rates are based on the liquid water content (lwc, in $m^3(aq)\,m^{-3}(air)$) whereas the backward exchange rates ($k_{ex}^{b}$) are based on the Henry's law coefficient ($H_s^{cp}$, in $mol\,(m^3\,Pa)^{-1}$), temperature (T, in K), and the universal gas constant (R, in $J\,(mol\,K)^{-1}$):

$$k_{ex}^{f} = k_{mt} \cdot lwc \tag{2}$$

$$k_{ex}^{b} = k_{mt} \cdot (H_s^{cp} \cdot R \cdot T)^{-1} \tag{3}$$

Here, $k_{mt}$ denotes the mass transfer coefficient of the given species. The mass transfer coefficient is limited by gas phase diffusion ($D_g$, in $m^2\,s^{-1}$) and is calculated for a single aerosol as:

$$k_{mt} = \left( \frac{r^2}{3D_g} + \frac{4r}{3\overline{v}\alpha} \right)^{-1} \tag{4}$$

where, r represents the particle radius (in m), $\alpha$ the accommodation coefficient of the given species, and $\overline{v}$ (in $m\,s^{-1}$) the mean molecular velocity from the Boltzmann velocity distribution.

### 3.3 Aqueous-phase mechanisms for AERCHEM

In AERCHEM, dissociation, hydration, and oxidation reaction rates are taken from the literature. The photolysis reaction rates are calculated outside AERCHEM and provided by the MESSy submodel JVAL (Sander et al., 2014). So far, all kinetic mechanisms used in MESSy submodels are build via the Kinetic PreProcessor (KPP; Sandu and Sander, 2006). To simplify the usage and enhance the consistency between all mechanisms used for the different phases (gas – MECCA, aqueous phase – SCAV, aerosol phase – GMXe–AERCHEM) the full mechanism is hosted within the MECCA submodel. Before compiling the MESSy code, the user is able to choose the required mechanisms. The supplemental material of this manuscript includes a manual for AERCHEM, outlining the procedure of selecting the desired mechanism. The following list provides a short overview about the tailor made aqueous-phase mechanisms currently available for AERCHEM, sorted by their complexity:





- The simplest aqueous-phase mechanism considers a few soluble compounds, their acid–base equilibria, and the oxidation of $SO_2$ by $O_3$ and $H_2O_2$ (abbreviated as Scm; Jöckel et al., 2006).

- A more complex aqueous-phase mechanism represents more than 150 reactions (abbreviated as Sc; Tost et al., 2007). It includes aqueous-phase $HO_x$ chemistry and the destruction of $O_3$ by $O_2^-$, but misses a detailed representation of aqueous-phase oxidation of oxygenated volatile organic compounds (OVOC). This mechanism can be considered the current standard mechanism for representing cloud chemical processes in MESSy (Jöckel et al., 2016).

- The most complex aqueous-phase mechanism is the recently developed Jülich Aqueous-phase Mechanism of Organic
Chemistry (abbreviated as JAMOC; Rosanka et al., 2021c, b, a). JAMOC includes a complex aqueous-phase OVOC oxidation scheme and represents the phase transfer of species containing up to 10 carbon atoms and the oxidation of species containing up to 4 carbon atoms. The photo-oxidation of species with 3 or 4 carbon atoms is limited to the major isoprene oxidation products (i.e. methylglyoxal, methacrolein, and methyl vinyl ketone) and the aqueous-phase sources of methylglyoxal. Overall, JAMOC represents the phase transfer of 350 species, 43 equilibria (acid–base and hydration),
and more than 280 photo-oxidation reactions. A detailed description of JAMOC is presented by Rosanka et al. (2021c). When using JAMOC, the user needs to select the Mainz Organic Mechanism (MOM) to represent gas-phase chemistry in MECCA.

A detailed comparison of all three mechanism is provided by Rosanka et al. (2021b, their Table 1). All reaction rates, Henry's law and accommodation coefficients, and other model parameters are provided by Rosanka et al. (2021c) and Sander (2021).

## 3.4 Solving the ODE system and numerical challenges

In order to numerically integrate the aqueous-phase chemical reaction mechanism, AERCHEM uses KPP. When using the KPP software, the user may select between several numerical solvers. For numerically complex multiphase chemistry problems, Rosenbrock solvers are known to be some of the most efficient solvers. Due to its favorable performance, the Rodas-3 (Sandu et al., 1997) Rosenbrock integrator, with automatic time step control, is selected as the default integrator in AERCHEM. We
find that Rodas-3 provides the best combination of efficiency and stability, when using relative and absolute tolerances of $1 \times 10^{-3}$ and $1 \, \mathrm{molecule \, cm^{-3}}$, respectively.

Due to the phase transfer reactions and equilibria, the stiffness of the ODE system increases with decreasing aerosol liquid water content. For this reason, AERCHEM performs the chemistry calculations only in the two larger hydrophilic (accumulation and coarse) modes in series. The accumulation mode is calculated first, conforming to the order utilized for ISORROPIA-II.
In order to ensure a proper stability of the numerical solver, AERCHEM is only executed if the aerosol liquid water content exceeds $10^{-14} \, \mathrm{m^3(aq) \, m^{-3}(air)}$. This low limit is two orders of magnitude lower than an earlier attempt to represent non-equilibrium aerosol chemistry on global scales by Kerkweg et al. (2007), who did not use operator splitting of the gas- and aqueous-phase chemistry. In their study, the non-equilibrium aerosol chemistry was almost exclusively executed in the marine boundary layer. With the limit used in AERCHEM, calculations of non-equilibrium aerosol chemistry are available over
continental regions.



## 4 Example results using AERCHEM

The primary objective of this section is to showcase initial findings obtained by using AERCHEM in GMXe within MESSy. Here, the fifth-generation European Centre Hamburg general circulation model (ECHAM5, version 5.3.02; Roeckner et al., 2003) is used as the core atmospheric model. This combination is known as the ECHAM/MESSy Atmospheric Chemistry

(EMAC) model. The physics subroutines of the original ECHAM code have been modularized and reimplemented as MESSy submodels and have continuously been further developed. Only the spectral transform core, the flux-form semi-Lagrangian large scale advection scheme, and the nudging routines for Newtonian relaxation are remaining from ECHAM. Our focus lies in determining whether AERCHEM adequately represents background concentrations rather than episodic events. As such, we restrict our comparisons to long-term observational datasets containing numerous observations at multiple sites and

exclude single measurement campaigns that are limited w.r.t. spatial and temporal representativeness. Examining the latter would require detailed process studies for specific conditions, which is beyond the scope of this study. For the comparison, we primarily emphasize inorganic aerosol mass concentrations, which are frequently observed.

### 4.1 EMAC modeling setup

In order to keep the computational demand low, we evaluate the implication of AERCHEM by applying EMAC at a resolution

of T42L31, i.e. with a spherical truncation of T42 (corresponding to a quadratic Gaussian grid of approximately 2.8° by 2.8° in latitude and longitude) with 31 vertical hybrid pressure levels up to 10 hPa of which about 22 levels represent the troposphere. Here, we use the standard time step length for this resolution of 900 s. In order to reproduce the actual day-to-day meteorology in the troposphere, the dynamics have been weakly nudged (Jöckel et al., 2006) towards the ERA-Interim (Dee et al., 2011) reanalysis data of the European Centre for Medium-Range Weather Forecasts (ECMWF).

Atmospheric gas-phase chemistry is represented in MECCA using the Mainz Organic Mechanism (MOM) recently evaluated by Pozzer et al. (2022). MOM contains an extensive oxidation scheme for isoprene (Taraborrelli et al., 2009, 2012; Nölscher et al., 2014; Novelli et al., 2020), monoterpenes (Hens et al., 2014), and aromatics (Cabrera-Perez et al., 2016; Taraborrelli et al., 2021). In addition, comprehensive reaction schemes are considered for the modelling of the chemistry of $NO_x$, $HO_x$, $CH_4$, and anthropogenic linear hydrocarbons. VOCs are oxidised by OH, $O_3$, and $NO_3$, whereas $RO_2$ reacts with $HO_2$,

$NO_x$, and $NO_3$ and undergoes self- and cross-reactions. All in all, MOM considers 43 primarily emitted VOCs and represents more than 600 species and 1600 reactions (Sander et al., 2019). In order to push EMAC to its technical limits, we represent the aqueous-phase chemistry in cloud droplets, rain (i.e., by using SCAV), and in deliquescent aerosols (i.e., by using AERCHEM) using JAMOC (see Sect. 3.3).

Anthropogenic emissions are based on the Emissions Database for Global Atmospheric Research (EDGAR, v4.3.2; Crippa

et al., 2018) and vertically distributed following (Pozzer et al., 2009). The Model of Emissions of Gases and Aerosols from Nature (MEGAN; Guenther et al., 2006) is used to calculate biogenic VOC emissions. Biomass burning emission fluxes are calculated using the MESSy submodel BIOBURN, which calculates these fluxes based on biomass burning emission factors and dry matter combustion rates. For the latter, Global Fire Assimilation System (GFAS) data are used, which are based





on satellite observations of fire radiative power from the Moderate Resolution Imaging Spectroradiometer (MODIS) satellite
instruments (Kaiser et al., 2012). GMXe considers the emission of $SO_2$ from anthropogenic activities (EDGAR, v4.3.2),
biomass burning (BIOBURN), and volcanic activities based on the AEROCOM dataset (Dentener et al., 2006). For primary
organic aerosol (POA) and black carbon (BC) emissions, GMXe considered anthropogenic emissions in the lower troposphere
and by aviation activities (EDGAR, v4.3.2), and from biomass burning (BIOBURN). Mineral dust emissions are calculated
online following Astitha et al. (2012) as bulk inert dust. Sea spray aerosol emissions are calculated online following Kerkweg
et al. (2006b), assuming the chemical composition proposed by Seinfeld and Pandis (2016, their Table 8.8). A summary of all
emissions considered in GMXe, including all related scaling factors, is provided in the Fortran Namelist S1 in the supplemental
material.

Within this study, we perform one simulation for 2010 using 2009 as spin up. This simulation was performed at the Jülich
Supercomputing Centre using the Jülich Wizard for European Leadership Science (JUWELS) cluster (Jülich Supercomputing
Centre, 2019).

## 4.2 Inorganic aerosol composition

In the following, EMAC simulated aerosol masses using AERCHEM for sulfate ($SO_4^{2-}$), nitrate ($NO_3^-$), ammonium ($NH_4^+$),
and chloride ($Cl^-$) are evaluated. We compare annual mean concentrations to three in situ monitoring networks: (1) for the
United States we rely on the Clean Air Status and Trends Network (CASTNET) operated by the Clean Air Markets Division
of the U.S. Environmental Protection Agency (EPA) providing weekly filter pack observations, (2) for Europe we use the co-
operative program for monitoring and evaluation of the long-range transmission of air pollutants in Europe (EMEP), and (3) for
East Asia we use the Acid Deposition Monitoring Network in East Asia (EANET). Observed concentrations are interpolated
onto the EMAC grid. If multiple stations coincide with the same model grid box, the average of all these stations is used
for the comparison. In addition, the aerosol composition simulated by ISORROPIA-II and AERCHEM are compared at each
observational site. Both compositions are obtained from the same EMAC simulation by providing the mass concentration of
each species simulated by ISORROPIA-II (which is used as an AERCHEM input) and by AERCHEM as separate model
outputs.

### 4.2.1 Sulfate ($SO_4^{2-}$)

Figure 2a shows the annual surface mean sulfate ($SO_4^{2-}$) concentration simulated by EMAC using AERCHEM and observed at
the three monitoring networks in 2010. Overall, the model reproduces the observed concentrations well. In the United States,
the model nicely captures the east-west as well as the north-south gradient in the sulfate EPA observations. The simulated
sulfate concentrations for almost all EPA stations are within a factor of two of the observed values (Fig. 2b). Only for two
stations, EMAC using AERCHEM predicts values that are slightly higher than a factor of two.

For an overwhelming number of EPA stations in the eastern US, the consideration of AERCHEM reduces EMAC's bias in
predicting sulfate compared to simulated values by ISORROPIA-II (indicated by down pointing triangles in Fig. 2). Further
reductions in the model bias are expected by accounting for the reactive uptake of IEPOX from isoprene which produces





stable organo sulfates (Eddingsaas et al., 2010; Wieser et al., 2023). An insignificant difference between simulated values of ISORROPIA-II and AERCHEM is observed in the Midwest. A similar good agreement is observed in Europe, where the east-west gradient is also nicely matched, even though EMAC tends to be biased low, especially in continental East Europe. Overall,

the model agrees reasonably well in Japan, South Korea, Russia and China, but tends to be biased high. However, in South East Asia, especially in Myanmar, Thailand, and in Kuala Lumpur in Malaysia, the model tends to significantly overpredict sulfate concentrations. These regions are highly photochemically active regions, where chemical sulfate losses might be of importance that have been recently described by Pan et al. (2019), Ren et al. (2021), Liu et al. (2021), and Cope et al. (2022). With the development of AERCHEM, all these processes can now be explicitly represented in EMAC to further reduce the observed

biases.

Some of the differences in the sulfate concentrations simulated by AERCHEM compared to simulated values from ISORROPIA-II, are related to the treatment of non-equilibrium conditions and kinetic limitations, considered in GMXe. GMXe first calculates the amount of each gas phase species considered in ISORROPIA-II that is kinetically able to condense onto the aerosol, by assuming diffusion limited condensation. This is achieved by extending the calculation for $H_2SO_4$ used in the M7 aerosol

model presented in Vignati et al. (2004) for $HNO_3$, $NH_3$, and HCl. Afterward, the thermodynamic equilibrium is calculated using ISORROPIA-II. In AERCHEM, the diffusion limitation is directly included in the calculation of the phase transfer reactions, i.e., the Henry equilibrium is corrected by the kinetic diffusion limitation, which can become highly relevant in case of further aqueous phase reactions of the dissolved compounds.

### 4.2.2  Nitrate ($NO_3^-$)

The annual mean nitrate ($NO_3^-$) concentrations simulated by EMAC using AERCHEM and observed at the EPA, EMEP, and EANET stations are shown in Fig. 2c. In the continental US, EMAC simulates the spatial pattern observed by the EPA network reasonably well. Large nitrate concentrations are simulated within a factor of two (Fig. 2d), but the model tends to overpredict low nitrate concentrations in the Midwest and Northeastern states. However, compared to the nitrate concentrations simulated by ISORROPIA-II, AERCHEM reduces EMAC's bias in simulated nitrate concentrations. Similar to the US, EMAC is biased

high in continental Europe, but the number of stations in Europe for which EMAC predicts nitrate concentrations higher than a factor of two is lower. The spatial variability with higher concentrations in Central Europe and lower concentrations in Northern-Europe is reasonably matched. EMAC tends to reproduce nitrate hotspots in the Benelux countries, as well as nitrate concentrations in Ireland. There is one significant outlier in Switzerland, where EMAC predicts significantly higher nitrate concentrations than observed. This station is located on the Jungfraujoch at about 3570 m. Due to the coarse model resolution

used, EMAC is not capable to resolve the high elevation of this station properly, leading to significantly higher simulated values. It is important to keep in mind that nitrate concentrations reported by EMEP are mainly based on Teflon filters and thus potentially systematically underestimated (Ames and Malm, 2001). In general, nitrate concentrations in South East Asia (e.g., Myanmar, Thailand, Vietnam, Cambodia, Malaysia, Indonesia) are properly simulated and nitrate hotspots in East Asia, like in central China or Jakarta, are reasonably well reproduced by EMAC. In Japan and South Korea, nitrate is slightly

overestimated but the usage of AERCHEM reduces EMAC's bias. In the remote marine boundary layer (i.e., on Okinawa




and at the Ogasawara Islands) EMAC tends to slightly overpredict nitrate concentrations. The improvements provided by AERCHEM may stem from the reaction of nitrate anion with the $SO_4^{2-}$ leading to nitrate radical which either outgasses or photolyses efficiently. Nevertheless, a much larger reduction of the model overpredictions are expected by including the known chemistry of reactive nitrogen essentially mediating $NO_x$-recycling via production of HONO (Ye et al., 2017; Andersen et al.,
2023) and $ClNO_2$ (Thornton et al., 2010). The role of particulate organic nitrate for predictions of inorganic nitrate is yet to be assessed. Many organic nitrates are known to hydrolize (Liu et al., 2012; Boyd et al., 2015; Vasquez et al., 2020) not always leading to a release of $NO_3^-$ (Zare et al., 2019).

### 4.2.3   Ammonium ($NH_4^+$)

In the eastern US, EMAC overall matches the EPA observations reasonably well but overestimates ammonium concentrations
in the Midwest when using AERCHEM (see Fig. 2e). For only four stations, the simulated difference in the concentration is slightly higher than a factor of two. Even though, EMAC is capable to represent the East-West gradient in the US, due to the slight overestimation in the Midwest the simulated east-west gradient is too low. The consideration of non-equilibrium aqueous-phase chemistry in aerosols leads to a reduced model bias for most EPA stations. Ammonium concentrations in Central Europe are reasonably well reproduced and almost all simulated values are within a factor of two. EMAC again fails to reproduce
low ammonium concentrations as observed at the Jungfraujoch (see discussion above). For this station, the lowest value in continental Europe is observed. Again, EMAC is capable to reproduce the North-South gradient in Europe, but underestimates its amplitude and thus systematically overestimates concentrations in Northern Europe. In East Asia, EMAC systematically overpredicts ammonium concentrations in continental regions, Japan, and the remote marine boundary layer but manages to reproduce ammonium hotspots (e.g., Central China, Central Java) reasonably well. Except for a few stations, simulated
ammonium concentrations in the US by AERCHEM are associated with a lower model bias compared to concentrations from ISORROPIA-II simulated for the same EMAC simulation. For EUROPE and East Asia, only minor differences are simulated.

### 4.2.4   Chloride ($Cl^-$)

Over the Central US, EMAC using AERCHEM tends to overestimate chloride ($Cl^-$) concentrations (Fig. 2g). More importantly, EMAC reproduces chloride concentrations in costal regions (e.g., Florida, San Francisco) frequently influenced by sea
salt emissions. Similarly, observations in Ireland, Island, costal regions in the Benelux countries, and coastal regions in Northern Europe are well captured by EMAC. High chloride concentrations in costal regions in East Asia are also well reproduced, especially in the remote marine boundary layer (i.e., on Okinawa and at the Ogasawara Islands). At a few observational stations in south Japan, EMAC tends to slightly underestimate very high chloride concentrations. Overall, differences in simulated concentration from AERCHEM and ISORROPIA-II at in-situ measurement stations are minor. In AERCHEM, chloride is not inert
and undergoes fast oxidation by hydroxyl radical triggering production of HCl, HOCl and $Cl_2$. The latter two are relatively insoluble and efficiently transfer chlorine to the gas phase (Soni et al., 2023; Dalton et al., 2023).





### 4.3 Aerosol acidity

#### 4.3.1 Aerosol acidity calculations

The aerosol $pH$ is defined as the negative decimal logarithm of the hydrogen ion activity ($a_{H^+}$):

$$pH = -\log_{10}(a_{H^+}) \tag{5}$$

where the hydrogen activity can be calculated by multiplying the hydrogen ion activity coefficient ($\gamma_{H^+}$) and the hydrogen ion molarity ($x_{H^+}$, in $mol\,L^{-1}$). In order to account for the differences induced by the non-equilibrium aerosol chemistry, we calculate the aerosol $pH$ for fine particles ($PM_{2.5}$, diameter $< 2.5$ µm) in order to allow for direct comparisons with observational data. For this, the hydrogen ion molarity is estimated by:

$$x_{H^+_{PM_{2.5}}} = \frac{\sum[H^+]_i \cdot f_{PM_{2.5_i}}}{\sum[H_2O]_i \cdot f_{PM_{2.5_i}}} \cdot \rho_{H_2O} = \frac{[H^+]_{PM_{2.5}}}{[H_2O]_{PM_{2.5}}} \cdot \rho_{H_2O} \tag{6}$$

where, $\rho_{H_2O}$ is the water density ($g\,L^{-1}$), and $[H^+]_i$ and $[H_2O]_i$ are the hydrogen ion mass concentration ($\mu g\,m^{-3} \equiv \mu mol\,m^{-3}$) and water mass concentration ($\mu g\,m^{-3}$) of the hydrophilic mode i, respectively. $f_{PM_{2.5_i}}$ represents the volume fraction of the given hydrophilic aerosol mode contained in fine particles with a diameter below $2.5$ µm. The $pH$ calculations are carried out exclusively when an adequate amount of water exists within the aerosol (total $PM_{2.5}$ water exceeds $0.01\,\mu g\,m^{-3}$). For 315 the $pH$ calculations for ISORROPIA-II and AERCHEM, we assume that the hydrogen ion activity coefficient is one. All $pH$ calculation are performed based on instantaneous output provided every five hours.

#### 4.3.2 Simulated aerosol acidity

Figure 3a and 3b shows the yearly mean aerosol $pH$ of fine particles ($PM_{2.5}$, diameter $< 2.5$ µm) based on the $H^+$ concentration simulated by ISORROPIA-II and AERCHEM, respectively. Separate model outputs for the $H^+$ concentration are provided after 320 the calculation performed by ISORROPIA-II and AERCHEM for the same EMAC simulation. In both cases, the aerosol liquid water content is calculated following Sect. 2.3. Here, the yearly mean fine aerosol $pH$ based on $n$, the number of five hourly model outputs per year, is calculated as:

$$\overline{pH_{PM_{2.5}}} = \frac{1}{n}\sum_{i=1}^{n} -\log_{10}\left(a_{H^+_{PM_{2.5_i}}}\right) \tag{7}$$

When non-equilibrium aerosol chemistry is not taken into account (i.e., simulated values from ISORROPIA-II), EMAC predicts 325 predominantly alkaline fine particles over the ocean. Further, mostly acid particles are simulated over continental regions influenced by anthropogenic activities. For continental regions in the Northern Hemisphere above 60°N and Australia, AERCHEM predicts slightly higher aerosol acidity. Differences in aerosol acidity simulated by AERCHEM compared to ISORROPIA-II in Central Europe and the Southeast US are only minor. In some polluted continental regions (e.g., China, South East Asia, Central Africa, Mexico, Central South America), on the other hand, the usage of AERCHEM results in slightly higher aerosol



pH compared to ISORROPIA-II predictions. Interestingly, these differences in fine aerosol acidity are not driven by the accumulation mode for which AERCHEM exclusively simulates a higher acidity over continental regions (see Fig. S1). They are rather driven by the slightly higher pH simulated for the coarse mode (see Fig. S2), which contributes only slightly to fine aerosols. In addition, AERCHEM predicts slightly more alkaline fine particles over major deserts (e.g., Sahara, Lut Desert, Thar Desert, and Arabian Desert). The most substantial differences in the aerosol acidity simulated are governed for fine particles

over the ocean. Exclusively higher fine aerosol acidity is simulated over all major oceans. At the same time, a high variability in differences between values simulated by AERCHEM and ISORROPIA-II is observed. The highest differences are simulated over the Southern Ocean, central Atlantic Ocean, and central Pacific Ocean. Lower differences are simulated over the Indian Ocean, northern and southern Atlantic, the southern Pacific, and in the northern Pacific just west of the US. Sea-salt aerosol particles are mainly emitted into the coarse mode. ISORROPIA-II simulates these aerosols to be alkaline (see

Fig. S2) whereas AERCHEM suggests a higher acidity. Acidification of sea-salt aerosols is partly due to the relatively fast oxidation of chloride by hydroxyl radical which eventually leads to hydrochloric acid formation. Moreover, methanesulfonic acid from DMS oxidation is a strong acid and contribute to lower the pH. As expected, this effect is more pronounced over photochemically active regions with high sea-salt and/or DMS emission.

### 4.3.3  Comparison to observational datasets

Evaluating the pH prediction skills of an atmospheric chemistry model is difficult since no direct measurements of aerosol acidity are available and observed aerosol acidity are estimated using thermodynamic equilibrium models (e.g., ISORROPIA-II). Assumptions made when using these models, ranging from the species that are considered (e.g., crustal species), over stable vs. metastable assumptions, to averaging over certain time periods, can significantly affect the predicted aerosol pH. In addition, the spatial variability is limited and mostly bound to continental regions in the Northern Hemisphere. Still, in order

to represent the spatial variability of aerosol acidity simulated by EMAC, we include pH values for fine aerosols derived from observations compiled by Pye et al. (2020) in Fig. 3a and 3b. However, please keep in mind that, due to the large uncertainties in observed aerosol acidity, these values are not intended to evaluate the model at a specific location. Overall, both ISORROPIA-II and AERCHEM reasonably reproduce aerosol acidity in the USA, Europe, Mexico, and South East Asia. In northern Asia, very high aerosol pHs are observed, which both AERCHEM and ISORROPIA-II fail to reproduce. By predicting a higher

acidity for fine aerosols in the marine boundary layer, EMAC's predictions skills seem to improve when using AERCHEM. This is especially true for observations made at the Guiana Basin, in the Southern Ocean north of Antarctica where the largest difference in pH is simulated, and observations south of Australia. The higher aerosol acidity in the marine boundary layer is in line with a recent measurement study by Angle et al. (2021), suggesting a fast acidification of sea spray aerosols within minutes.





## 5 Model limitations

### 5.1 Thermodynamic activity

In highly concentrated solutions, non-ideal behavior can occur. To account for these conditions within thermodynamic models (e.g., ISORROPIA-II), thermodynamic activities are considered in calculating thermodynamic equilibrium. As discussed by Pye et al. (2020), the assumptions and actual activities of inorganic compounds considered in different thermodynamic models vary significantly, while predicting activity coefficients for organic compounds remains challenging due to limited measured values. In the current version of AERCHEM, we do not account for thermodynamic activities. Estimating the effect of ignoring these is difficult, given the high uncertainty in activity coefficients. Most thermodynamic equilibrium models do not consider organic compounds; however, one exception is the Aerosol Inorganic–Organic Mixtures Functional groups Activity Coefficient (AIOMFAC; Zuend et al., 2008, 2011) model. This model predicts thermodynamic activity coefficients for liquid mixtures containing water, inorganic ions, and organic compounds. AIOMFAC covers a wide variety of organic compounds by applying a group-contribution approach considering a set of organic functional groups. The incorporation of AIOMFAC into AERCHEM would allow predicting the thermodynamic activity of each compound represented in each of the aqueous-phase mechanisms available in AERCHEM.

### 5.2 Ionic strength

High ionic strength can lead to "salting in" or "salting out" effects that influence the Henry's law solubility constants of certain species. This effect is assessed and calculated by determining the Sechenov constant, which typically does not change the solubility in pure water by more than an order of magnitude (Yu and Yu, 2013). Recent studies have highlighted the significance of ionic strength on the partitioning of ambient water-soluble organic gases within cloud, fog, and aerosol water. Pratap et al. (2021) demonstrated that sulfate salts can induce "salting in" or "salting out" effects, while chloride salts always result in "salting out" effects. Monovalent cations, on the other hand, exhibit no significant salting effect. Additionally, reaction rate constants may be influenced by the ionic strength of the solvent, although kinetic data in this area is limited (Herrmann et al., 2015; Mekic and Gligorovski, 2021). In order to properly represent the phase transfer, a representation of salting effects in SCAV and AERCHEM is planned in the future.

### 5.3 Crustal elements

In a recent study, Karydis et al. (2021) showed that crustal elements such as $K^+$, $Mg^{2+}$, $Ca^{2+}$ from dust and $Na^+$ from sea salt reduce the aerosol acidity in certain regions. As global atmospheric chemistry models often disregard these elements, they may produce biased low predictions of aerosol pH. When using ISORROPIA-II within GMXe to perform thermodynamic calculations, the model considers crustal elements. However, these elements are not incorporated into any of the mechanisms employed by AERCHEM to represent non-equilibrium aerosol chemistry. Consequently, when using AERCHEM, the simulated pH may be biased low and could potentially impact the partitioning between the gas phase and deliquescent aerosols.



Karydis et al. (2021) demonstrated that the effect of crustal elements is globally minor. Ideally, crustal elements should be taken into account by AERCHEM. However, developing a comprehensive representation of crustal elements in the kinetic model is beyond the scope of this study. On the other hand, crustal elements always dissolve together with specific anions depending on the dust mineralogy. For dust emissions, the assignment of the anions associated with crustal elements is critical for the impact 395 on acidity as the associated cations are only very weak Lewis acids.

## 6 Future applications

The advancement of AERCHEM enables exploring an extensive range of novel subjects. The following list highlights a selection of topics that the MESSy community intends to investigate using AERCHEM in the foreseeable future:

1. Recent research conducted by Kluge et al. (2023) provides an extensive dataset of vertical profiles and total vertical
column densities of glyoxal (CHOCHO) in the troposphere using an airborne mini-DOAS onboard the German High Altitude and LOng Range (HALO) research aircraft. Their study focused on various atmospheric conditions, including pristine terrestrial, pristine marine, mixed polluted, and biomass burning affected air masses. Kluge et al. (2023) compared each flight campaign to an EMAC simulation using an extensive gas-phase oxidation scheme for isoprene, monoterpenes, and aromatics and identified discrepancies between the model's simulated and observational data in dif-
ferent environments. EMAC tends to underpredict glyoxal vertical profiles and total vertical column densities in marine environments (e.g., Mediterranean Sea, East China Sea, Tropical Atlantic). In contrast to marine environments, EMAC tends to overpredict glyoxal vertical profiles and total vertical column densities in biogenic dominated regions (e.g., Amazon rainforest). This discrepancy may be due to the model neglecting the uptake of glyoxal in cloud droplets and deliquescent aerosols, which is known to compete with photochemical losses of glyoxal (e.g., Volkamer et al., 2007;
Kim et al., 2022). By incorporating detailed OVOC aqueous-phase chemistry (i.e., JAMOC) in cloud droplets and deliquescent aerosols (i.e., by using AERCHEM), the representation of glyoxal will be improved, allowing to establish an updated global glyoxal budget.

2. Previous modelling studies have emphasized the critical role of aqueous-phase oxidation processes in shaping secondary organic aerosol (SOA) formation (e.g., Carlton et al., 2010). Although JAMOC currently incorporates oligomerization
reactions for glyoxal and methylglyoxal, the resulting tracers are not considered as SOA products. This limitation may result in an underestimation of global SOA formation in EMAC, which currently does not account for aqueous-phase production within cloud droplets or deliquescent aerosols. With the implementation of AERCHEM, we can now overcome these technical constraints and explicitly represent SOA formation arising from aqueous-phase processes. This expansion is not limited to glyoxal and methylglyoxal but may also encompass other precursors like isoprene-epoxydiols (IEPOX),
recently developed for the MESSy submodel MECCA by Wieser et al. (2023). By incorporating these advanced representations, MESSy gains improved accuracy and comprehensiveness in capturing atmospheric SOA formation.



3. The representation of aqueous-phase chemistry in EMAC is significantly influenced by proper accounting for oxidants. In particular, Fenton chemistry (Deguillaume et al., 2004) plays an essential role in generating OH. Several highly idealized box-model studies (e.g., Mouchel-Vallon et al., 2017) have demonstrated the importance of this OH production mechanism. This suggests that EMAC may currently underestimate the impact of aqueous-phase chemistry in regions with high concentrations of iron (Fe), such as the Sahara, Lut Desert, Thar Desert, and Arabian Desert. In addition to these areas, Central Africa - characterized by substantial biogenic VOC emissions - may also be influenced by Fe transported from the Sahara. Furthermore, mineral dust is frequently transported across the tropical Atlantic to reach the Amazon Basin. Representing iron solubility in global models poses significant challenges and requires careful consideration of various simplifications and assumptions. For instance, some approaches rely on simplified representations of oxalate ($C_2O_4^{2-}$), such as discussed by Hamilton et al. (2019). By incorporating an advanced iron dissolution scheme (e.g., Ito and Xu, 2014; Ito, 2015) into the chemical mechanisms utilized by AERCHEM, we can now calculate iron solubility online based on calculated aerosol pH and oxalate concentrations. Integrating this enhanced representation of iron solubility within EMAC's chemical mechanisms allows for a more comprehensive assessment of the importance of Fenton chemistry in global aqueous-phase processes. This, in turn, enables us to better understand and quantify the impact of Fe transport on atmospheric aerosol formation and associated climate feedbacks. A prominent example is the recently reported evidence on the interaction between sea-salt and mineral dust and its impact on the atmospheric oxidation capacity (van Herpen et al., 2023).

4. Radical chemistry in polluted environments heavily affected by burning of fossil fuel and/or biomass is not well understood yet. For instance, efficient formation mechanisms for HONO are still elusive. Although the particle-phase photolysis of nitrate has been proposed to be important (Ye et al., 2017; Andersen et al., 2023), observational constraints on aged biomass burning plumes indicate the need to revisit the relevant chemistry (Peng et al., 2022). In addition, high levels of chloride in continental urban air masses have been reported and shown to enhance radical production by interacting with reactive nitrogen at night (Thornton et al., 2010). However, model studies are usually limited to the representation of relevant chemistry by using surface reactions uptake coefficients with little dependence on aerosol composition. In AERCHEM, key reactions for the production of HONO, $ClNO_2$ and $Cl_2$ can now be investigated by incorporating the recent advancements on the multiphase kinetics of chlorine (Soni et al., 2023; Dalton et al., 2023).

## 7 Conclusions

This manuscript introduces the development of the AERosol CHEMistry (AERCHEM) sub-submodel, version 1.0, integrated as an add-on to the thermodynamic equilibrium model of the MESSy submodel GMXE and shows first results obtained with the atmospheric chemistry model EMAC. Its ability to represent non-equilibrium aqueous-phase chemistry with varying levels of complexity for deliquescent fine aerosols is a novelty among Chemistry Climate Models (CCMs) available worldwide. To demonstrate the capabilities of AERCHEM, we compared simulated values with observational data from three in situ monitoring networks. The comparison revealed that AERCHEM captures background concentrations of sulfate, nitrate, ammonium,



and chloride ions reasonably well. Especially in the US, incorporating non-equilibrium aqueous-phase chemistry into the model led to reduced modelling biases of sulfur, nitrate, and ammonium when compared with simulated concentration based on GMXe's thermodynamic equilibrium model. In most cases, AERCHEM simulates too high chloride mass concentrations over continental regions, but more importantly reproduces concentrations in costal regions and the marine boundary layer. However, compared to simulated chloride values from the thermodynamic equilibrium model, the usage of AERCHEM does

not result in a significant model bias reduction. Although the usage of AERCHEM results in only minor differences in aerosol acidity over continental regions, it simulates significantly higher acidity for fine aerosols in the marine boundary layer, which is consistent with observations and literature.

The improved representation of aerosol acidity by AERCHEM has great potential to enhances MESSy's capabilities to realistically simulate air quality, aerosol toxicity, acid deposition, and aerosol cloud interactions. In particular, over oceanic regions,

we anticipate substantial differences in cloud condensation nuclei (CCN) activation that could have far-reaching implications on cloud properties and thus climate. AERCHEM enables investigations of the global-scale impact of aerosol non-equilibrium chemistry on atmospheric composition. In the future, by exploring key multiphase processes, AERCHEM contributes to improved model predictions for oxidation capacity and aerosol distribution in the troposphere. This in turn leads to improved understanding of chemistry-climate interactions, resulting in more accurate climate projections and better informed policy

decisions related to air quality management.

*Code and data availability.*  The Modular Earth Submodel System (MESSy, doi:10.5281/zenodo.8360186) is continuously further developed and applied by a consortium of institutions. The usage of MESSy and access to the source code is licenced to all affiliates of institutions which are members of the MESSy Consortium. Institutions can become a member of the MESSy Consortium by signing the MESSy Memorandum of Understanding. More information can be found on the MESSy Consortium Website (http://www.messy-interface.org,

last access: 24 August 2023). The code presented/used here (doi:10.5281/zenodo.10036115) has been based on MESSy version 2.55.2 (doi:10.5281/zenodo.8360276) and will be part of the next official release.

The model outputs relevant for this study are permanently stored in the Zenodo repository, accessible through doi:10.5281/zenodo.10059700. The EPA CASTNET, EMEP, and EANET datasets can be downloaded from https://www.epa.gov/castnet (last access: 22 August 2023), https://ebas.nilu.no/ (last access: 22 August 2023), and https://monitoring.eanet.asia/document/public/index (last access: 22 August 2023),

respectively. The observed global fine aerosol acidity dataset can be downloaded from doi:10.23719/1504059 .

*Author contributions.*  SR, DT, and HT were responsible for the conceptualisation of this study. SR and HT developed and reviewed the technical realisation of AERCHEM. SR, HT, DT, RS, PJ, and AK performed all necessary technical modifications in MESSy to implement AERCHEM. SR performed the numerical simulations and wrote the manuscript. All authors contributed to reviewing and editing of the final manuscript.



*Competing interests.* The contact author has declared that neither he nor his co-authors have any competing interests, besides the fact that several co-authors are executive (AK, RS) or topical (HT, PJ) editors for GMD.

*Acknowledgements.* The authors gratefully acknowledge the invaluable contributions made by Prof. Astrid Kiendler-Scharr, who sadly passed away in early 2023. Her expert guidance and insightful advice played a crucial role in shaping the early discussions and development of this project.

The authors gratefully acknowledge the Earth System Modelling Project (ESM) for funding this work by providing computing time on the ESM partition of the supercomputer JUWELS at the Jülich Supercomputing Centre (JSC).



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

755





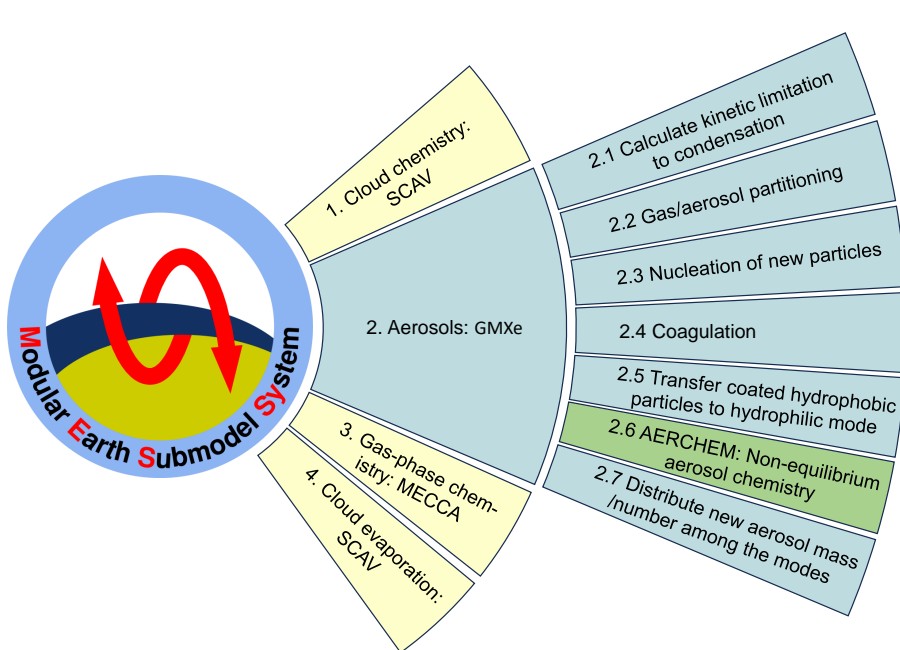

**Figure 1.** Graphic summary of the calling sequence of chemical processes within MESSy (left) and the calling sequence of processes in the GMXe submodel (right).





**Figure 2.** Annual surface mean for (a) sulfate ($SO_4^{2-}$), (c) nitrate ($NO_3^-$), (e) ammonium ($NH_4^+$), and (g) chloride ($Cl^-$) concentrations simulated by EMAC using AERCHEM for the year 2010. Annual surface mean observational concentration for stations in the EPA (USA), EMEP (Europe), and EANET (East Asia) network are depicted as triangles and boxes. A triangle pointing down indicates a model bias reduction when using AERCHEM compared to ISORROPIA-II, whereas a triangle pointing up indicates a model bias increase. Boxes indicate station for which no difference between AERCHEM and ISORROPIA-II is simulated. Panels (b), (d), (f), and (h) show the direct comparison between model simulated values and observations from EPA, EMEP, and EANET for sulfate, nitrate, ammonium, and chloride concentrations, respectively.

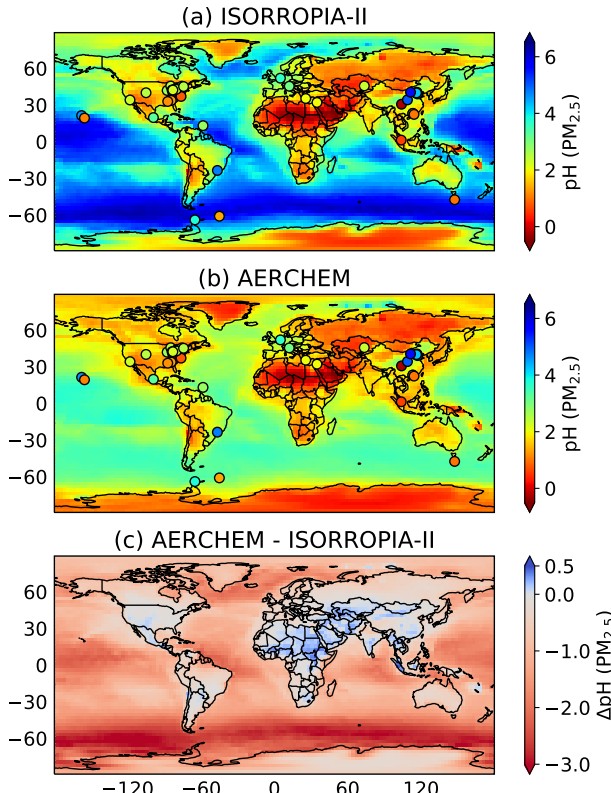

**Figure 3.** Mean yearly aerosol pH for fine particles ($PM_{2.5}$, diameter < 2.5 μm) simulated by (a) ISORROPIA-II and by (b) AERCHEM. Subfigure (c) represents the absolute difference of the yearly means. In both cases, the aerosol liquid water content is calculated following Sect. 2.3. Section 4.3.1 elaborates on how the aerosol pH for fine particle is calculated based on the four hydrophilic lognormal modes. The yearly mean is calculated following Eq. (7). Please note that for the figure showing the absolute pH differences, an increase in acidity (decrease in pH) is indicated by red shading, whereas an increase in pH is indicated in blue. For comparison, observed fine particle acidity, based on the dataset published by Pye (2020), is indicated by circles in panels (a) and (b).





**Table 1.** GMXe submodel setup used in this study. In GMXe, aerosol species are distributed between the 4 hydrophilic and 3 hydrophobic aerosol modes. Table adapted from Pringle et al. (2010).

| Mode | Abbr. | $R_P$ | $H_2O$ | $SO_4^{2-}$ | $NO_3^-$ | $Cl^-$ | $NH_4^+$ | $Na^+$ | BC | Du | SS | POC | SOA | AERCHEM |
|---|---|---|---|---|---|---|---|---|---|---|---|---|---|---|
| Hydrophilic (soluble) | | | | | | | | | | | | | | |
| Nucleation | NS | < 5 | P | P | P | | P | | | | | | | |
| Aitken | KS | 5 - 50 | P | P | P | | P | | P | | E | E | P | |
| Accumulation | AS | 50 - 700 | P | P | P | E | P | E | P | P | E | P | P | JAMOC |
| Coarse | CS | > 700 | P | P | P | E | P | E | P | P | E | P | P | JAMOC |
| Hydrophobic (insoluble) | | | | | | | | | | | | | | |
| Aitken | KI | 5 - 50 | | | | | | | E | | | E | P | |
| Accumulation | AI | 50 - 700 | | | | | | | | E | | | | |
| Coarse | CI | > 700 | | | | | | | | E | | | | |

$R_P$ = particle radius (nm), P = Permitted in the mode, E = Emitted into the mode, BC = black carbon, Du = dust, SS = sea spray, POC = primary organic carbon, SOA = secondary organic aerosols