# Peer review of "How non-equilibrium aerosol chemistry impacts particle acidity: the GMXe AERosol CHEMistry (GMXe–AERCHEM, v1.0) sub-submodel of MESSy"

_EGUsphere, 2023_

## Author Comment (AC1)

We thank both reviewers for their insightful and beneficial feedback. Throughout this document, any remarks provided by the reviewers are indicated by black text, while our corresponding replies are highlighted in red.

**Reply to comments of Anonymous Referee #1**

This manuscript provides a description of a newly developed sub-submodel module, AERCHEM, designed to enable representation of non-equilibrium aqueous-phase chemistry within the Modular Earth Submodel System (MESSy) Global Modal-aerosol eXtension (GMXe). The first half of the manuscript explains the how AERCHEM functions, its components, and its place within the larger submodel, while the second half of the manuscript benchmarks the model output by comparing it to observations and equilibrium-based simulations of aerosol inorganic components and pH. The manuscript is well written and provides a helpful while still brief summary of how AERCHEM works. However, the usefulness of the second half of the manuscript related to model performance is severely limited by the lack of specifics, as described in more detail below.

Thank you very much for your helpful comments and seeing the potential in our work. We strongly agree that the second half of the manuscript needs to be more specific. In order to address this, we made major changes to this part of the manuscript. Further details are elaborated below.

Most notably, indicators of overall model bias and, more importantly, the change in model bias between ISORROPIA and AERCHEM are entirely qualitative, not quantitative. Aside from what's visible (but still lacks quotable numbers) in Figure 2 and mentions in the manuscript of where the model-measurement deviation exceeds a factor of two, all comparisons between simulated and observed values or values between two different simulations are rendered in broad, general terms like "reproduces observations reasonably well", "reduces the model bias" [without numbers], "similar to the predictions of ISORROPIA" (taken just from lines 10-13). Researchers interested in using this submodel will not be satisfied with such generalities; to make this useful to future users, model biases and uncertainties for each of the analyzed model outputs (or at least the inorganic aerosol composition, since pH observations come with so much uncertainty that their quantification is always a bit of a guess) should be quantified and clearly reported. What, for example, was the normalized mean bias of simulated sulfate relative to global observations with ISORROPIA, and how much did that change with AERCHEM?

We completely agree with the reviewer that the current evaluation is currently too qualitative and further evaluation is needed. Keeping this in mind, we extended our model evaluation and added a fourth figure (now Fig. 3) that shows Taylor diagrams for each inorganic species. The evaluation is performed for each region (observation network) and each season, for both ISORROPIA-II and AERCHEM. To allow the comparison between different observational networks as well as different seasons in one Taylor diagram, we normalized the standard deviation by the observed standard deviation. We extended the discussion in Sect. 4.2 accordingly. Due to the nature of the pH dataset (different measurement techniques used, different observation years, etc.), we refrain from performing this analysis for the pH dataset. Thus, the Taylor diagrams are only added for the inorganic species.

Furthermore, most model users will want to focus on specific regions and times for such purposes as comparing simulated loadings of aerosol species to measurements. To that end, It would be particularly

helpful to quantify the model bias for each species by region, rather than the general discussion presently in the manuscript. For nitrate in particular, strongly temperature-dependent partitioning means that biases could be very different season to season, and it would be highly useful to see this broken down more; however, such model outputs may not exist from the single runs performed here, and it may not be worth running a whole additional simulation just to get this new seasonal breakdown.

We fully agree with the reviewer that a separate statistical analysis for each region and season is warranted. Therefore, we added the evaluation (see above comment) for each observational network and thus region (USA, Europe, and Asia) and for each season (DJF, MAM, JJA, and SON) in addition to the full year analysis.

The most useful, but perhaps most difficult, addition of quantitative information to the manuscript would be to include concrete numbers of how much specific processes contribute to the changes between ISORROPIA and AERCHEM outcomes in the model. For example, how much of the higher acidity of sea-salt particles in AERCHEM is contributed by chloride + OH oxidation, how much from methanesulfonic acid, and how much from other pathways? This may also not be possible to calculate from existing model output and considered beyond the scope of the current manuscript, but I would encourage the authors to revisit the statements made throughout the manuscript on attribution of changes and provide as much quantitative detail as possible.

It would be ideal to perform a complete budget analysis of all species included in JAMOC (or at least the inorganic species evaluated in the manuscript). In the past, such an analysis has been frequently used with EMAC's gas-phase chemistry submodel (Gromov et al., 2010). Recently, the same methodology was expanded to be available in SCAV and theoretically in AERCHEM. However, a model description and evaluation of this technique for aqueous phase processes deserves a dedicated manuscript which is beyond the scope of our study.

Further more minor comments are accompanied by line numbers referring to their position in the manuscript.

L 14 - "significant" should be "significantly"

Done.

L 67-69 - this isn't a sentence; "distributions. Three" should either be turned into one sentence by replacing the period with a colon, or an independent clause should be added to the fragment.

Done.

L 261 - higher than a factor of two relative to what.

In this statement we refer to the comparison between AERCHEM and the observational network (EMEP). We rephrased this statement to the following: "Similar to the US, EMAC is biased high in continental Europe, but the number of stations in Europe for which AERCHEM predicts nitrate concentrations higher than a factor of two compared to observations from EMEP, is lower."

L 263-266 - despite the issues with coarse model resolution, can the model comparison with the Jungfraujoch station tell us anything about free tropospheric aerosol composition and how well it'l simulated?

Due to the combination of the coarse horizontal resolution and the limited number of layers (here 31) we are very careful with such claims. But we agree that it would be very interesting to evaluate how vertical aerosol profiles change when using AERCHEM. However, such an evaluation should be performed with the appropriate horizontal and vertical resolution and is thus beyond the scope of this manuscript. By using the MESSy submodel SORBIT and S4D (Jöckel et al. 2010), the model could be evaluated with ATOM campaign data and e.g., CALIPSO observations.

L 273-274 - This sentence isn't clear. Are you saying that the HONO and ClNO2 production are included in AERCHEM but aren't producing as big a model reduction in nitrate as you'd expect? Or that future updates including these reactions would reduce model overpredictions even more?

This statement discusses a potential process that might result in further changes when using AERCHEM. However, at the moment this process is not included in JAMOC but we are planning to include it in the future. We updated the statement for clarification as follows: "Nevertheless, a much larger reduction of the model overpredictions are expected by including the known chemistry of reactive nitrogen essentially mediating NOx-recycling via production of HONO (Ye et al., 2017; Andersen et al., 2023) and ClNO2 (Thornton et al., 2010), which is currently not included in JAMOC."

L 275-276 - Similar question for this sentence -- are these particular organic nitrate hydrolysis reactions included in AERCHEM or not?

Like the previous comment, these reactions are not included in AERCHEM at the moment. Recently, Wieser et al. 2023 added the hydrolysis of isoprene nitrates, which will be available in AERCHEM once their manuscript is published. We plan to add the hydrolysis of more organic nitrates in the future.

We added the following statement for clarification: "Even though these processes are currently not included in JAMOC, a global analysis of the importance of organic nitrate hydrolysis reactions can be easily realized, due to the flexible design of AERCHEM."

L 281 - no comma needed after "even though"

Done.

L 281 and 286 - "capable to" should be "capable of"

Done.

Section 4.2.3 - To what extent are the differences between ISORROPIA and AERCHEM ammonia just a response to changes in sulfate and nitrate, versus specific facets of new ammonium chemistry?

This is an interesting aspect. At the moment, JAMOC only represents the uptake of ammonia and its protonation. It is thus highly likely that changes in ammonium are related to the changes in sulfate and nitrate. We cannot currently address this fully without a proper budget analysis between ISORROPIA-II and AERCHEM. We added the following statement to section 4.2.3.: "At the moment, JAMOC only represents the uptake of ammonia and its protonation. Thus, the changes in ammonium are potentially mainly related to the changes in sulfate and nitrate. A proper budget analysis, like the methodology presented by (Gromov et al., 2010) is thus warranted in the future."

L 293-294 - Why is this more important? (same issue on L 458).

Due to the coarse spatial resolution used and the limited continental emissions compared to oceanic emissions, matching coastal sites was more important to us during the development of AERCHEM. We agree that these statements are misleading and thus removed them.

L 295 - Does "Island" refer to Iceland?

Yes. Thank you for spotting this. We changed it accordingly.

L 296 - "costal" should be "coastal"

Done.

L 298-301 - it is surprising that despite these important differences in chlorine chemistry between the two models, the two aerosol modules give similar results for aerosol chloride content (although you haven't quantitatively told us how similar). Does this mean that the hydroxyl radical initiated oxidation of chlorine to insoluble species is unimportant, or is it offset by additional sources?

The rate constant is high but likely the low concentration of OH is not consuming much of the chloride, which is very abundant. Writing "fast oxidation" is thus misleading. We expect that missing reactions following $N_2O_5$ uptake and $NO_3^-$ photolysis are likely to have a larger impact. We updated the manuscript accordingly.

L 315 - to what extent could this assumption of a unity activity coefficient be biasing results? While it's understandable (as you write in Section 5.1) that the difficulty of estimating activity coefficients means you don't bother to implement them here, it is worth at least some discussion of what effect that might have on results.

It is difficult to estimate to which extent this assumption influences the predicted pH values. It is worth noting that ISORROPIA-II also assumes unity for the activity coefficient of $OH^-$ and $H^+$. Fountoukis et al. 2007 state that: "$\gamma OH^-$ and $\gamma H^+$ are assumed equal to unity, as the activity coefficient routines cannot explicitly calculate them." Also assuming unity for AERCHEM, allows for a fair comparison between both models. However, compared to AERCHEM, ISORROPIA-II estimates the activity coefficient for all other inorganic species, which surely indirectly affects the prediction of $H^+$. To which extent is difficult to estimate, without major code modifications to ISORROPIA-II.

L 332 - why does the coarse mode contribute at all to fine mode acidity? Aren't the coarse and fine mode two separate bins? Overall, the discussion of what drives differences in aerosol acidity is confusing, complicated in part by fact that different terms related to acidity (pH, acidity, and alkalinity) are all being intercompared and seemingly used interchangeably.

You are right that GMXe uses multiple modes to represent different size bins. In this simulation we use four (see Table 1) different soluble modes (nucleation, Aitken, accumulation, and coarse). Here, the coarse mode represents all aerosols with a diameter above 1.4 μm. For Sect. 4.3 we define fine particles with a diameter below 2.5 μm to allows some limited comparison with observational data. Thus, a fraction of the coarse size mode may be part of fine particle cutoff. To account for this, we take the fractional contribution of the coarse aerosol mode into account when calculating aerosol acidity (see Eq. 6 and 7 in the revised manuscript). To reduce the potential confusion, we rephrase the discussion to the following: "Interestingly, for the accumulation mode, AERCHEM simulates a higher acidity over continental regions (see Fig. S2) but tends to simulate slightly higher pH for the coarse mode (see Fig. S3). This suggests that even though the

coarse mode (particles diameter > 1.4 µm) only contributes minor fractions to the fine aerosol acidity, changes in the fine aerosol pH are driven by coarse mode compositional changes."

L 334 - "governed" doesn't seem to fit here. Is this sentence just meant to say that fine particles over the ocean are the category for which simulated pH is most different between the two models?

Agreed. We changed the sentence to: "The most substantial differences in aerosol acidity are simulated for fine particles in the marine boundary layer."

L 463 - "enhances" should be "enhance"

Done.

Figure 2 caption: "Boxes indicate station for which no difference" --> "station" should be "stations", and within what margin is "no difference" calculated? [as an aside, this figure is very pretty!]

Thank you very much for spotting this. We define a station that with "no difference" as a station where the yearly mean difference between AERCHEM and ISORROPIA-II does not exceed 5%. We updated the caption accordingly.

SI section 1, first paragraph: "xylens" should be "xylene", or maybe "xylenes" if you're referring to multiple isomers

We updated it to xylenes.

**References**

Fountoukis, C. and Nenes, A.: ISORROPIA II: a computationally efficient thermodynamic equilibrium model for K+–Ca2+–Mg2+–NH4+–Na+–SO42−–NO3−–Cl−–H2O aerosols, Atmos. Chem. Phys., 7, 4639–4659, https://doi.org/10.5194/acp-7-4639-2007, 2007.

Gromov, S., Jöckel, P., Sander, R., and Brenninkmeijer, C. A. M.: A kinetic chemistry tagging technique and its application to modelling the stable isotopic composition of atmospheric trace gases, Geosci. Model Dev., 3, 337–364, https://doi.org/10.5194/gmd-3-337-2010, 2010.

Jöckel, P., Kerkweg, A., Pozzer, A., Sander, R., Tost, H., Riede, H., Baumgaertner, A., Gromov, S., and Kern, B.: Development cycle 2 of the Modular Earth Submodel System (MESSy2), Geosci. Model Dev., 3, 717–752, https://doi.org/10.5194/gmd-3-717-2010, 2010.

Wieser, F., Sander, R., and Taraborrelli, D.: Development of a multiphase chemical mechanism to improve secondary organic aerosol formation in CAABA/MECCA (version 4.5.6-rc.1), Geosci. Model Dev. Discuss. [preprint], https://doi.org/10.5194/gmd-2023-102, in review, 2023.

---

## Author Comment (AC2)

We thank both reviewers for their insightful and beneficial feedback. Throughout this document, any remarks provided by the reviewers are indicated by black text, while our corresponding replies are highlighted in red.

**Reply to comments of Anonymous Referee #2**

The manuscript presents the new development of AERCHEM for the representation of non-equilibrium aqueous phase chemical reactions, as an addition to the thermodynamic equilibrium model ISORROPIA-II, in Earth System modelling. The manuscript is well organized by first presenting the different submodels for treatment of atmospheric aerosols, and after that presenting an application of AERCHEM in global simulations of the inorganic aerosol composition including a detailed evaluation against measurements from three monitoring networks. Further, the acidity of aerosols is compared in terms of pH to limited observations of a global dataset. The manuscript is interesting both from a practical viewpoint of mechanism development and from a scientific viewpoint given the relevance of aerosol-cloud interactions for climate. My main concern is the incomplete description of the connection between AERCHEM and the thermodynamic equilibrium computation. The abstract and text describes AERCHEM as an add-on to ISORROPIA-II, meaning that AERCHEM calculations are done in series with the thermodynamic equilibrium calculations. It remains unclear which variables are transferred from ISORROPIA to AERCHEM and what exactly constitutes the difference in the simulations. The manuscript should be revised according to the specific comments and technical remarks below.

Thank you very much for your helpful comments and seeing the potential in our work. We agree that further elaborations on the variable transfer between ISORROPIA-II and AERCHEM is needed. We changed the manuscript accordingly. Further details are elaborated below.

Specific Comments:

1.) Please add a section with the description of the coupling AERCHEM – ISORROPIA in GMXe. For example, it seems like aerosol water content is first calculated in ISORROPIA, and that the aerosol water after adding the water uptake of organic constituents is then used as reaction volume for the non-equilibrium reactions in AERCHEM. Further, it is mentioned on page 9 that GMXe first calculates the amount of each gas phase species that is kinetically able to condense onto the aerosols using the aerosol model M7. Then the equilibrium partitioning of gases to the liquid phase happens in ISORROPIA. How is it avoided that this affects the uptake of gases afterwards in AERCHEM? There is already some explanation on page 9, which should be further extended to get the complete picture of the coupling (see point 5 below). Suggest to create an additional schematic illustration of the program flow that illustrates the transfer of variables between the two submodels.

Just to clarify, the liquid water content (LWC) used in AERCHEM is the sum of the inorganic LWC calculated by ISORROPIA-II and the organic LWC calculated as described in Sect. 2.3. The gas phase concentrations transferred to AERCHEM also include the fraction of the gas phase concentrations that was calculated to not condense onto the aerosol by GMXe before executing ISORROPIA-II. This is necessary, since the diffusion limit is considered in the phase transfer calculations included in AERCHEM.

We agree that a figure summarizing the data transfer between ISORROPIA-II and AERCHEM is a great addition to the manuscript. We thus added a flow chart illustrating this as a second panel to Fig. 1. In addition, we added an extensive elaboration on the data transfer between both models to Sect. 3.1.

2.) The effect of crustal elements (like potassium) is considered in ISORROPIA, but not in AERCHEM. Does this mean that the difference between AERCHEM and ISORROPIA simulations is (a) the non-equilibrium aqueous phase reactions and (b) the omission of crustal elements associated with dust emissions and biomass burning? The crustal elements do not only increase aerosol pH but also increase nitrate formation, for example, dust aerosols that contain calcium may react with nitric acid to form calcium nitrate, which significantly contributes to nitrate concentrations when dust emission and industrial emissions coincide in a grid cell. In this regard, it would be illuminating to perform one simulation with EMAC excluding the crustal elements considered in ISORROPIA.

As stated in Sect. 4.1, the mineral dust is emitted as bulk inert dust. This means that no crustal elements are emitted. In addition, we list all aerosol emissions from biomass burning that exclude all crustal elements that are emitted by biomass burning activities. This means that the above proposed simulation exactly represents the simulation performed in this study.

To avoid any confusion, we changed the statement to the following: "Mineral dust emissions are calculated online following Astitha et al. (2012) as bulk inert dust, i.e., no crustal elements are emitted."

3.) I strongly recommended to include a comprehensive graphic panel for the presentation of the comparison of model simulations to observations of the inorganic aerosol composition, showing box-and-whisker plots (min, max, median, 25th percentile, 75th percentile) of observations, AERCHEM, and ISORROPIA. One plot per sulfate, nitrate, ammonium, chloride where each plot includes all observation stations of one monitoring network. This totals to 4 x 3 plots, fitting on two pages.

We agree that this is a great addition to the manuscript. However, our intent was to keep the manuscript as compact as possible (as recognized by the first referee). Therefore, we decided to add these box plots to the supplemental material.

4.) Section 4.2.1 (Sulfate): Please add information on how much sulfate is produced in clouds compared to the sulfate produced by gas-to-particle conversion and aerosol aqueous phase production.

With the output from the simulation that we performed, we cannot calculate the amount of sulfate that is produced by cloud process nor by gas-to-particle conversion. In the past, budget analyses have been frequently performed with EMAC's focusing on gas-phase processes (Gromov et al., 2010). Recently, the same methodology was expanded to be available in SCAV and theoretically in AERCHEM. However, a model description and evaluation of this technique for aqueous phase processes deserves a dedicated manuscript and is beyond our study.

5.) The paragraph on page 9, starting with "Some of the differences .:." could be used in the explanation of the connection between ISORROPIA and AERCHEM.

Thank you for this hint. We moved this discussion part to Sect. 3.1 in which we discuss the data transfer between ISORROPIA-II and AERCHEM.

6.) Section 4.2.2 (Nitrate): In several places of this section, EMAC simulations are referred without mentioning whether this was EMAC using either AERCHEM or ISORROPIA or rather EMAC using

ISORROPIA. Maybe first state in which world regions only marginal differences were found between the two submodels and then state where the use of AERCHEM results in differences.

Just to clarify, we only perform one simulation for which we created separate outputs of the aerosol composition predicted by ISORROPIA-II and AERCHEM (as stated in Sect. 4.2).

To avoid confusion that multiple simulations were performed, the statement in Sect. 4.2 now reads: "Both compositions are obtained from the same EMAC simulation by providing the mass concentration of each species simulated by ISORROPIA-II (which is used as an AERCHEM input) and by AERCHEM as separate model outputs. The exact location where both compositional information are obtained in GMXe is summarized in Fig. 1b." Additionally, we revised this section to clarify when we are discussing results form ISORROPIA-II or AERCHEM.

7.) P10, Line 279-280: Is the overestimation of ammonium concentrations in the Midwest US connected to the overestimation of low nitrate concentrations?

This is a great point.

8.) P12, Line 333-335: It is a bit difficult to understand why fine particle over major deserts simulated with AERCHEM are slightly more alkaline, given that the crustal elements are not incorporated in the aqueous phase chemistry mechanisms of AERCHEM (P13, Line 388-389) but only in the thermodynamic calculations.

As mentioned in our response to your comment 2, no crustal elements from mineral dust are considered by ISORROPIA-II or AERCHEM in the simulation performed. Therefore, the slightly higher aerosol pH over desert regions is not a result of a different treatment of crustal elements across both models. The change in pH is a result of the compositional changes predicted by AERCHEM.

Technical Corrections:

P4, L 101: what is "cloud species"?

In this context, cloud species refers to all aqueous phases species that are dissolved in the cloud droplets of a given grid box. Since this is a rather technical reference, we changed "cloud species" to "cloud tracer".

P4, L 101: should "GMEx" be replaced by "GMXe"?

Done.

P4, Line 101-103: "After GMEx and MECCA have calculated all aerosol processes and gas-phase chemistry …"; does this refer to the non-activated particles? Are the aerosol operators and chemistry operators running during the cloud periods?

If a cloud droplet is present during the integration time step, the fraction of the activated aerosol will reside in the cloud droplets during the gas phase and aerosol calculation. The activated fraction of the aerosol, i.e., the material dissolved in the cloud phase does not take part in any gas phase or aerosol phase chemical or microphysical processing. The initial composition considered in GMXe thus includes only the non-activated particles. If a cloud is present, aerosol and gas phase processes are calculated. However, MESSy relies on operator splitting, meaning that these processes are calculated in sequence (see Fig. 1a).

P12, Line 340-341: "oxidation of chloride by hydroxyl radical"; please provide a global map of the hydroxyl radical concentration in coarse mode.

With the current output that is available from the simulation performed in this study we are not able to provide a global map of the hydroxyl radical concentration for the coarse or the accumulation mode. The usage of AERCHEM comes at high computational cost. Due to the limited computational time available in our computing projects, rerunning the two-year simulation just to create this plot is unfeasible. We prefer to allocate this time for the further developments proposed in Sect. 6.

P12, Line 345 and Line 355: Rather not refer to "prediction skill" when the comparison to observed aerosol acidity should be qualitative, "better agreement with observations" is more adequate here. Please give the mean pH for observations and models for coastal and marine environments.

We totally agree with the reviewer's comment. We changed it to "agreement with observations". The mean pH across all coastal regions are: observation – 2.7, ISORROPIA – 1.8, AERCHEM – 1.8; and across all observation locations in the marine boundary layer are: observation – 2.1, ISORROPIA – 3.9, AERCHEM – 2.6. We added these values to Sect. 4.3.3.

P14, Line 394-396: "For dust emissions, the assignment of the anions associated with crustal elements is critical for the impact on acidity as the associated cations are only very weak Lewis acids." Do you mean "associated anions are only very weak Lewis acids", such as carbonates and silicates? Please rephrase sentence, avoid using "associated" twice in the sentence.

Yes, we meant carbonates and sulfates. We have reformulated the sentence as below:

"As the cations of the crustal elements are only very weak Lewis acids, the simulated impact of dust emissions on acidity critically depends on the assignment of the fraction of anions (sulfate, carbonate, or hydroxide) that are emitted along."

P15, Line 431-433: It should be noted that the presence of titanium in iron-containing mineral dust might enhance iron dissolution from mineral dust. Further, nitric and sulphuric acids will interact with other metal cations in the mineral dust and have a synergistic effect on overall iron mobilization (Hettiarachchi et al., 2018, https://doi.org/10.1021/acs.jpca.7b11320). Ilmenite could be a good proxy for the complexity of iron-containing mineral dust.

Thank you very much for this useful hint. We will keep this in mind in the future development of AERCHEM to allow the representation of iron solubility.

We added the following statement to the manuscript: "Further, these approaches do not take into account that the presence of titanium in iron containing mineral dust might enhance iron solubility or that the presence of sulfuric and nitric acid in mineral dust will interact with other metal cations affecting iron mobilization (Hettiarachchi et al., 2018)."

Figure 1: ISORROPIA is not depicted in the slices for GMXe.

The calculations of ISORROPIA-II are performed in 2.2 of Figure 1. We updated the text of box 2.2 accordingly.

**References**

Gromov, S., Jöckel, P., Sander, R., and Brenninkmeijer, C. A. M.: A kinetic chemistry tagging technique and its application to modelling the stable isotopic composition of atmospheric trace gases, Geosci. Model Dev., 3, 337–364, https://doi.org/10.5194/gmd-3-337-2010, 2010.